# Study of Water Distribution, Textural and Colour Properties of Cold Formulated and Air-Dried Apple Snacks

**DOI:** 10.3390/foods11050731

**Published:** 2022-03-02

**Authors:** Juan Manuel Castagnini, Silvia Tappi, Urszula Tylewicz, Luca Laghi, Pietro Rocculi

**Affiliations:** 1Department of Agricultural and Food Sciences, University of Bologna, Piazza Goidanich 60, 47521 Cesena, Italy; juan.castagnini@unibo.it (J.M.C.); silvia.tappi2@unibo.it (S.T.); l.laghi@unibo.it (L.L.); pietro.rocculi3@unibo.it (P.R.); 2Interdepartmental Centre for Agri-Food Industrial Research, University of Bologna, Via Quinto Bucci 336, 47521 Cesena, Italy

**Keywords:** vacuum impregnation, functional compounds, fruits, blueberry juice, TD-NMR

## Abstract

Vacuum impregnation is considered a cold formulation technology since it allows the incorporation of a desired functional compound into porous plant tissue without applying any heat. It is widely used in combination with the drying process to obtain added-value snacks. The aim of this work was to evaluate the effect of two trehalose concentrations (5 and 10% *w*/*w*) on: (i) the water state and texture evolution during the air drying (50 °C, 8 h) of apple snacks vacuum impregnated with blueberry juice, and on (ii) the colour of the final dried apple snacks. The results of nuclear magnetic resonance (NMR) showed that trehalose affects the water mobility of the samples during drying especially after 200–300 min of drying. In terms of textural properties, trehalose could increase the crispier characteristic of the samples impregnated with trehalose at the end of drying. Significative changes were found in terms of chroma and hue angle.

## 1. Introduction

Food-related consumer behaviour is a complex phenomenon, which could be associated with different meanings of food, such as sacred, moral, health, social, and aesthetic [1]. More recently, Kokkoris and Stravrova [2] observed that health-conscious behaviours (e.g., following a healthy diet, frequent consumption of fruits and vegetables, etc.) are not only driven by the health meaning of food but are strongly driven also by social and moral ones. Therefore, the consumption of fruit-based healthy snacks has reported a growing trend around the world. Even during the COVID-19 pandemic, a study conducted on the Italian population during the first lockdown in 2020 [3] indicated an increase in healthy food consumption and in the involvement in cooking and a decrease in the consumption of junk food.

Vacuum impregnation (VI) followed by air drying at low temperature is a common technology to obtain shelf-stable value-added products [4,5,6,7]. Firstly, VI is used to replace the air in porous plant tissue with the solution containing functional compounds [8,9,10,11]. Secondly, the so obtained products are stabilized by drying to ensure a long shelf-life at room temperature and to provide them with the desired physicochemical characteristics (e.g., colour and texture) which make them attractive for consumers [12,13]. While during the VI of fruits their moisture content is not modified, it is important to reduce it during drying to the desired level, hence, it is important to strictly control the endpoint of the drying process that is crucial to maintain a consistent product quality [14,15].

Therefore, the development of a real-time online detection device for moisture content could be very important in order to ensure product quality; however, besides the water content, for the control of the drying process it is very important to know the water state of the materials. The water state can be measured by time domain nuclear magnetic resonance (TD-NMR), which is a non-destructive rapid detection technology developed on the basis of the proton relaxation times–T2 [16,17]. According to the two-sites chemical exchange model proposed by Hills and Duce [18], the T2 of the protons located in different cellular compartments of the tissue is the weighted average between the T2 of water (around 2 s) and the T2 of the sites of the biopolymers it chemically exchanges with (in the order of milliseconds). 

This technology has been already applied by different authors for monitoring the redistribution of water through the substructures of the cells during the osmotic dehydration process of different fruits [19,20,21,22,23] as well as for the detection of water dynamics in apple cubes during microwave vacuum drying [24]. NMR/MRI detection systems have been used for the measurement of moisture content, moisture distribution, and the state of water in corn kernels during microwave vacuum drying [25]. In addition, Sun et al. [14,26] proposed the use of a non-destructive technique based on combined TD-NMR and artificial neural network technology (ANN) to monitor the moisture content in fruit and vegetables during microwave vacuum drying. 

However, little data is available concerning the effect of combined treatments such as vacuum impregnation and air-drying on the water state and physicochemical parameters of fruit. Therefore, the aim of this work was to evaluate the effect of two trehalose concentrations (5 and 10% *w*/*w*) on: (i) the water state and texture evolution during the air drying (50 °C, 8 h) of apple snacks vacuum impregnated with blueberry juice, and (ii) the colour of the final dried apple snacks.

## 2. Materials and Methods

### 2.1. Raw Material

Apples (var. Granny Smith) purchased from a local market in Cesena (Italy) were stored at 4 ± 1 °C and 80% relative humidity before use for not longer than 2 weeks. The moisture content of the fresh apples was 84 ± 1%. The apples were washed, peeled, and cut into 5 mm thick discs with the external and internal diameters of 60 mm and 23 mm, respectively.

Blueberry juice was obtained by crushing the undersized and overripe blueberries with a food processor (Model 27700-56, Russel Hobbs, England). An enzymatic depectinization and filtration was carried out as described in Castagnini et al. (2017) [27].

The trehalose dihydrate was purchased from ACEF S.P.A. (Piacenza, Italy).

### 2.2. Vacuum Impregnation

Vacuum impregnation (VI) of apples was carried out at 25 ± 1 °C with 3 different solutions: blueberry juice (VI control) and blueberry juice with the addition of 5 and 10% *w*/*w* of trehalose (VI 5 and VI 10, respectively). The samples were kept in the solutions at 200 mbar (absolute pressure) for 10 min, followed by 10 min at atmospheric pressure as reported by Betoret et al. (2007) [28]. Three replicates of the process were carried out for each type of sample.

### 2.3. Air-Drying

All prepared samples were air-dried at 50 °C for 8 h in a hot air cabinet dryer (POL-EKO-APRATURA SP.J., PL). The air velocity was set to 2 m/s, while the air renewal fee was set to 50%. The sample weight was recorded every 15 min for the first 2 h and every 30 min till the end of drying. The nal moisture content was 21.0 ± 0.6%, 9 ± 1%, and 10.4 ± 0.7%, for the VI control, VI 5 and VI 10, respectively. Three replicates of the process were carried out for each type of sample.

### 2.4. Analytical Determinations

#### 2.4.1. TD-NMR

The transverse relaxation time (T2) of protons was measured at 25 °C with the Carr–Purcell–Meiboom–Gill (CPMG) pulse sequence, using a Bruker Minispec PC/20 spectrometer (Bruker, Germany) working at 20 Hz. Apple cylinders of about 250 mg (h = 10 mm, d = 8 mm) were cut with a core borer and then inserted into the NMR tubes. The exponential decay comprised 32,000 echoes, a 180–180 echo time (τ) of 0.160 ms, leading to a dead time of 167 µs, and a recycle delay of 5 s. The τ value was kept below 500 ms, to separately observe the water populations in different cell compartments [29]. An overall impression of the relaxation times of the protons populations was obtained by means of the UPEN software [30], which inverts the T2-weighted signals using a quasi-continuous distribution of exponential curves and through fittings to the sum of an increasing number of exponential curves. Furthermore, a multi-exponential discrete fitting was successively applied, in agreement with similar works on fruits [20,29,31] to accurately determine the T2 and relative intensities of the water populations. Analyses were carried out in triplicate for each sample.

#### 2.4.2. Texture Analysis

A puncture test was carried out with a universal texture analyser mod. TA.HD (Stable Microsystems, Godalming, UK) equipped with a 25 kg load cell and a 2 mm diameter cylinder probe (P/2). The test speed and the trigger force were 2.0 mm/s and 0.04903 N, respectively. All samples were punctured to 100% distance, and at least 10 measurements were carried out for each sample. Force vs. distance curves were analysed through a macro developed with the software Exponent v. 6.1.20 (Stable Micro Systems) that calculates the maximum force (N) as the maximum absolute force at the sample rupture point, the slope (N/mm) as the slope of the line between the origin of coordinates and the sample rupture point and finally, the area (N/mm) below the curve from the origin of coordinates till the sample rupture point.

#### 2.4.3. Computer Vision System (CVS)

The colour of each sample was determined by means of a computer vision system (CVS) consisting of an illumination source (four daylight fluorescent lamps (60 cm in length) with a colour temperature of 6500 K), a colour digital camera (Mod. D7000, Nikon, Japan) with a 60 mm lens (Mod. AF-S Micro f/2.6G ED, Nikkor, Singapore, Japan) and an image processing software. The samples were placed inside a dark box to exclude external light. Three replicates of each sample were analysed.

The pre-processing of the RGB images, segmentation and colour analysis was performed with a Python script developed by the authors. The CIE L*a*b* colour parameters were calculated using the function rgb2lab from the colour module of the scikit-image package for Python.

### 2.5. Statistical Analysis

Significant differences between the samples were calculated using one-way analysis of variance with a *p*-level < 0.05, and the post-hoc Tukey test. The analysis was performed using the software STATISTICA 6.0 (Statsoft Inc., Bedford, UK).

## 3. Results and Discussion

### 3.1. Water State

Fruit and vegetables contain water in different forms such as free, immobile and bound water. According to Li et al. [24], free water, with high mobility, is present in vacuoles and in the extracellular compartment. Immobile water is present in the cytoplasm, and it is bound to the macromolecules and therefore trapped within highly organized structures, while bound water is present in the cell wall, since it is combined with cell wall polysaccharides in plant tissue. The attribution of water proton pools to different cell compartments could be obtained by measuring the transverse relaxation time-T2 and amplitude or relative intensity of the signal [14,20,21,26]. In the present work, three different water proton pools were identified in impregnated apples and attributed to the vacuoles (T2–1000–1300 ms), cytoplasm and extracellular spaces (T2-200–300 ms) and cell wall (T2-40–50 ms) according to our previous studies [20,31]. Similar findings were reported by Li et al. (2018) [24] in fresh apple tissue where three protons populations were found at approximately 11, 126, and 1335 ms. 

Figure 1 shows the relationship between the drying time and T2 of three differently impregnated apple samples in the cell wall (Figure 1a), cytoplasm/extracellular space (Cyt/ES) (Figure 1b) and vacuole (Figure 1c). A decrease of T2 was observed in all three water populations during drying, indicating a progressive loss of water mobility in each of them. The decrease of T2 from the protons populations located in the vacuole, cytoplasm and extracellular space was observed also by other authors; however, they stated that the bound water combined with cell wall polysaccharides was stable and not susceptible to the drying process [14,24,26]. Concerning the T2 of the proton pools in the cell wall (Figure 1a) and Cyt/ES (Figure 1b), there were no significant differences among the three samples, except for the 300 min, where the samples impregnated with the juice containing a higher amount of trehalose (VI 10) presented a significantly lower (*p* < 0.05) water mobility. As for the vacuole (Figure 1c), this trend was observed starting from 200 min of drying and it was maintained till the end of the process. The lower water mobility of the VI 10 samples could be due to the higher accumulation of trehalose in the tissue, thus increasing the glass transition temperature (Tg), as reported by Xin et al. (2013) [32]. 

Figure 2 shows the relative signal intensity evolution in the three cell compartments and moisture rate (MR) changes during the drying of the VI control (Figure 2a), VI 5 (Figure 1b) and VI 10 (Figure 1c) apple samples. In all the samples, a similar trend was observed. With the increase of drying time, a reduction of the MR was observed, as discussed in detail by Castagnini et al. (2021) [12]. In their work, the MR reduction during the drying of apple samples impregnated with trehalose-added blueberry juice was faster when compared to the control sample, making the drying process more sustainable. As it can be seen from Figure 2, the reduction of the MR was followed by the decrease of the relative intensity of proton pools in the vacuole. This indicates that the bulk water was more active and was removed first during the drying process. The relative intensity of the proton pools located in the cell wall and Cyt/ES increased with an increase of the drying time; however, at the end of the process, a slight increase for the Cyt/ES was observed in apples impregnated just with blueberry juice (Figure 2a), while in the samples with added trehalose, the trend remained increasing till the end of the process, probably due to the accumulation of the trehalose which could trap the water molecules (Figure 2b,c).

### 3.2. Texture during Drying

The evolution of the textural properties, such as maximum force (N), slope (N/mm) and area (N.mm) during drying is presented in Figure 3a–c, respectively. The maximum force, which is an index of the product resistance [33], remained fairly constant during the first half of the drying process. Then, between 300–330 min, it started to increase until the end of the drying time. In the case of the slope, which indicates the rigidity of the sample [34], the behaviour was quite different: in the beginning, the values were high, then, from 150 to 300 min, they decreased until a minimum and lastly, they increased again until the end of the dying when they returned to the initial level. Statistically significative differences were found between the control and the samples impregnated with trehalose (*p* < 0.05). In combination, the product resistance and the rigidity of the sample were related to a crispier characteristic. At the beginning of the drying, the behaviour of the maximum force and slope was determined by the moisture content or turgor pressure of the surface cells, and both intracellular and extracellular water, available in the surface layer in this period of drying, contributed to the viscoelastic behaviour of the apple slices [35]. Then, when the samples began to lose water, driven by the water diffusion and water convection from the surface [35], the structure softened. At the end of the drying, the samples turned crispier as a consequence of the low water content, the more compact structure as a result of the shrinkage, the collapse of cells and pores [36] and changes in the state of carbohydrates [37]. 

Finally, the value of the area, which is an index of energy required to break the structure, increased during the drying process. Similar results were also found in terms of hardness and compression work for osmodehydrated-candied pumpkins air-dried at different temperatures [37]. No statistically significant differences were found between the control and the trehalose-impregnated samples.

### 3.3. Colour Analysis

Colour is one of the most important quality indices of foods, especially in dried ones, both for their commercial value and for consumer choice [36], but the drying process can cause many changes in the colour of the samples. In the case of apples, the changes of colour during drying are usually caused by oxidative reactions that are associated with the drying temperature and the duration of the process [38]. In our case, however, the presence of blueberry juice masked the colour changes, which could be due to oxidation of the apple.

In Table 1 the colour parameters of each sample before the drying and after 480 min of drying are reported. In terms of luminosity (L*), statistically significant differences were observed among the groups of samples, but no differences were found for the same sample before and after drying. Instead, for the chroma (C*), that is the purity of the colour, statistically significant differences were found between almost all samples; however, a clear trend of the effect of drying and/or trehalose on this parameter was not present. Finally, some changes of hue (h*) could be observed, in particular, the value was always increasing after the drying process.

## 4. Conclusions

This study presents interesting information related to the evolution of the water state, texture and colour characteristics of vacuum impregnated apple slices during drying. 

A decrease of T2 was observed in all three water populations during drying, indicating a progressive loss of water mobility in each of them, which was more emphasized in the samples impregnated with trehalose. Indeed, the presence of the sugar, allowed a faster reduction of moisture in the last part of the drying process, due to its ability to trap water. 

In terms of textural properties, trehalose seemed to increase the crispier characteristic of the samples impregnated with trehalose at the end of the drying. Significative changes were found in terms of the sample colours especially for the chroma and hue angle.

## Figures and Tables

**Figure 1 foods-11-00731-f001:**
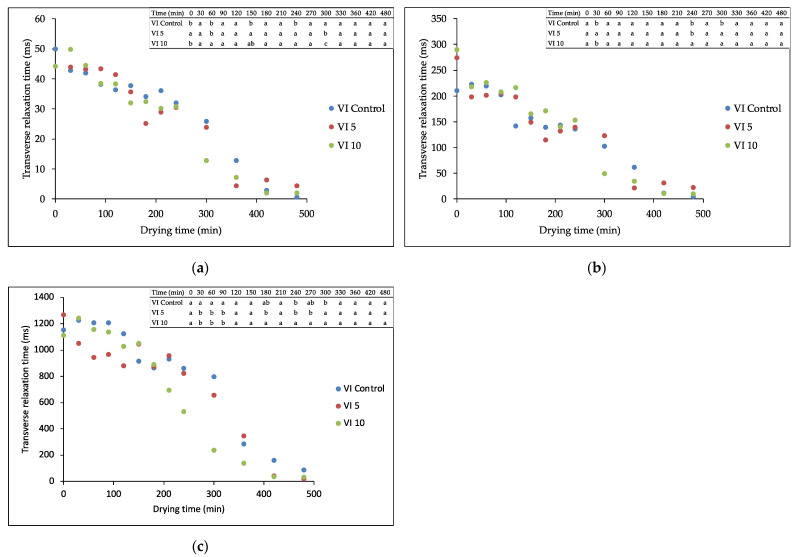
Transverse relaxation time evolution during drying for: (**a**) cell wall population; (**b**) cytoplasm/extracellular space (Cytoplasm/ES); and (**c**) vacuole.

**Figure 2 foods-11-00731-f002:**
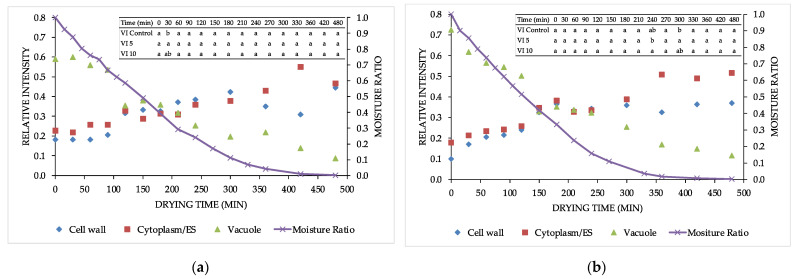
Relative signal intensity evolution and moisture rate changes during drying for: (**a**) VI control sample; (**b**) VI 5 sample; and (**c**) VI 10 sample.

**Figure 3 foods-11-00731-f003:**
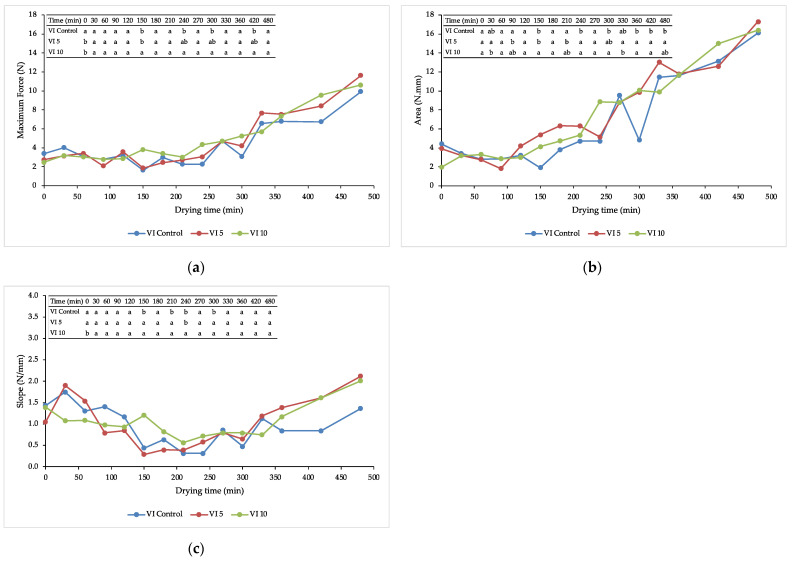
Changes in textural parameters during air-drying: (**a**) changes in maximum force at the rupture point; (**b**) changes in the slope of the line till the rupture point; and (**c**) changes in the area below the curve till the rupture point.

**Table 1 foods-11-00731-t001:** Colour parameters in the CIE L*a*b* space.

Sample	Drying Time (min)	L*	A*	B*	C*	H*
VI control	0	65 ± 4 ^c^	26 ± 4 ^e^	17 ± 3 ^b^	31 ± 4 ^d^	33 ± 3 ^a^
	480	64 ± 3 ^c^	21 ± 2 ^cd^	17 ± 1 ^b^	27 ± 2 ^bc^	40 ± 3 ^b^
VI 5	0	49 ± 3 ^a^	22 ± 3 ^d^	19 ± 3 ^b^	29 ± 4 ^c^	40 ± 3 ^b^
	480	50 ± 4 ^a^	19 ± 1 ^c^	17 ± 1 ^ab^	25.2 ± 0.9 ^b^	42 ± 4 ^b^
VI 10	0	58 ± 3 ^b^	10.9 ± 0.8 ^a^	15 ± 1 ^a^	19 ± 1 ^a^	54 ± 3 ^c^
	480	56 ± 4 ^b^	16 ± 3 ^b^	28 ± 4 ^c^	33 ± 3 ^d^	60 ± 8 ^d^

Different lowercase letters in columns indicate significant differences (*p* < 0.05) between the samples.

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
