# Peer review of "Study of Water Distribution, Textural and Colour Properties of Cold Formulated and Air-Dried Apple Snacks"

_foods, 2022, doi:10.3390/foods11050731_

Round 1
Reviewer 1 Report
The presented paper has demonstrated a great idea for obtaining healthy apple snack products. Considering that there is not much literature about combined treatments of vacuum impregnation and air-drying on fruit's water state and physicochemical parameters, this paper has a lot of potentials. However, after reading the manuscript, there is an impression of insufficient science depth. Even if the paper's subject is monitoring the water status, texture, and color, it would be more comprehensive if the authors had also provided information about the obtained product's chemical composition, antioxidant capacity, sensorial analysis, and shelf life.
Specific comments:
Is there any other reason the authors have chosen to use trehalose solutions other than increasing the crispiness of the samples?
What was the purpose of vacuum impregnation in the blueberry juice? Just for masking the oxidation of the apple samples or for nutritional enrichment of the snack product? Elaborate.
Page 2, line 70, add the moisture content of the fresh apple.
Page 2, line 92, add the moisture content of the snack product.
Thoroughly check the English through the paper; there are many mistakes.
Author Response
The presented paper has demonstrated a great idea for obtaining healthy apple snack products. Considering that there is not much literature about combined treatments of vacuum impregnation and air-drying on fruit's water state and physicochemical parameters, this paper has a lot of potentials. However, after reading the manuscript, there is an impression of insufficient science depth. Even if the paper's subject is monitoring the water status, texture, and color, it would be more comprehensive if the authors had also provided information about the obtained product's chemical composition, antioxidant capacity, sensorial analysis, and shelf life.
Specific comments:
Is there any other reason the authors have chosen to use trehalose solutions other than increasing the crispiness of the samples?
In this article we present the results of the trehalose addition especially related to water state and physical properties of apple snacks. We investigate the effect of trehalose on the anthocyanin content in another article already published:
Castagnini, J. M., Tappi, S., Tylewicz, U., Romani, S., Rocculi, P., & Dalla Rosa, M. (2021). Sustainable Development of Apple Snack Formulated with Blueberry Juice and Trehalose. Sustainability, 13(16), 9204. https://doi.org/10.3390/su13169204
What was the purpose of vacuum impregnation in the blueberry juice? Just for masking the oxidation of the apple samples or for nutritional enrichment of the snack product? Elaborate.
The purpose of vacuum impregnation with blueberry juice was to produce a snack enriched in antioxidant compounds and visually appealing. As stated in the previous answer. In this article we are publishing results related to the water state, colour, and texture properties.
Page 2, line 70, add the moisture content of the fresh apple.
The moisture content was added as requested by the revisor.
Page 2, line 92, add the moisture content of the snack product.
The moisture content was added as requested by the reviewer.
Thoroughly check the English through the paper; there are many mistakes.
The English was check through the whole paper.
Reviewer 2 Report
Manuscrpit "Study of water distribution, textural and color properties of cold formulated and air-dried apple snacks" (Castagnini et al.) Presents the results of research on the influence of the preparation process on the physical properties of apple snacks. The aim of this work was to evaluate the effect of trehalose concentrations and juice impregnation on the physical properties and color of snacks.
I ask the authors to consider making changes to the manuscript:
The abstract may be rewritten to give more test results.
The aim of the research given in the abstract differs from that given in the text, please provide the same purpose of the manuscript.
Figure 1, 2 and 3 - Was a statistical analysis of the results performed?
Table 1 - please standardize the approximations of the results.
In the chapter describing the effect of trechalose and juice impregnation, the authors observed differences in color parameters, while in the conclusions they wrote that these differences were not significant. Please explain.
Please, the authors write down how many repetitions the determinations were made for each sample.
Author Response
Manuscrpit "Study of water distribution, textural and color properties of cold formulated and air-dried apple snacks" (Castagnini et al.) Presents the results of research on the influence of the preparation process on the physical properties of apple snacks. The aim of this work was to evaluate the effect of trehalose concentrations and juice impregnation on the physical properties and color of snacks.
I ask the authors to consider making changes to the manuscript:
The abstract may be rewritten to give more test results.
The authors appreciate the opinion of the reviewer, but we consider that the abstract contains the most relevant information of the study.
The aim of the research given in the abstract differs from that given in the text, please provide the same purpose of the manuscript.
The aim of the research was checked and corrected as requested by the reviewer
Figure 1, 2 and 3 - Was a statistical analysis of the results performed?
One-way Anova was added to the figures.
Table 1 - please standardize the approximations of the results.
The approximations were calculated based on the significant figures of each mean and standard deviation. Because of that, there are results with different numbers of decimals.
In the chapter describing the effect of trehalose and juice impregnation, the authors observed differences in color parameters, while in the conclusions they wrote that these differences were not significant. Please explain.
The sentence on the conclusion was corrected.
Please, the authors write down how many repetitions the determinations were made for each sample.
The repetitions of each sample were written down.
Round 2
Reviewer 1 Report
The Author's response is fair; the paper can be accepted in present form